# An integrated approach to the characterization of immune repertoires using AIMS: An Automated Immune Molecule Separator

**Christopher T. Boughter**◯*, **Martin Meier-Schellersheim**◯*

Computational Biology Section, Laboratory of Immune System Biology, National Institute of Allergy and Infectious Diseases, National Institutes of Health, Bethesda, Maryland, United States of America

* christopher.boughter@NIH.gov (CTB); mms@NIAID.NIH.gov (MMS)

## Abstract

The adaptive immune system employs an array of receptors designed to respond with high specificity to pathogens or molecular aberrations faced by the host organism. Binding of these receptors to molecular fragments—collectively referred to as antigens—initiates immune responses. These antigenic targets are recognized in their native state on the surfaces of pathogens by antibodies, whereas T cell receptors (TCR) recognize processed antigens as short peptides, presented on major histocompatibility complex (MHC) molecules. Recent research has led to a wealth of immune repertoire data that are key to interrogating the nature of these molecular interactions. However, existing tools for the analysis of these large datasets typically focus on molecular sets of a single type, forcing researchers to separately analyze strongly coupled sequences of interacting molecules. Here, we introduce a software package for the integrated analysis of immune repertoire data, capable of identifying distinct biophysical differences in isolated TCR, MHC, peptide, antibody, and antigen sequence data. This integrated analytical approach allows for direct comparisons across immune repertoire subsets and provides a starting point for the identification of key interaction hotspots in complementary receptor-antigen pairs. The software (AIMS—Automated Immune Molecule Separator) is freely available as an open access package in GUI or command-line form.

## Author summary

Over the past decade, the success of immunotherapeutics coupled with the declining costs of sequencing have stimulated a near exponential growth in the identification of novel T cell receptor, peptide, and antibody sequences for use in combating disease and dysregulation. With these new datasets freely available to researchers, a wealth of analytical tools have been created for standardized data analysis. However, these tools are largely fragmented, capable of processing only singular molecular species, likewise generating fragmented interpretations of complex adaptive immune environments. In this manuscript,

**Data Availability Statement:** All data and code used for the analysis in this manuscript are freely available online with no restrictions. All input FASTA sequences and code needed to recreate the

analysis can be found via the AIMS GitHub page: github.com/ctboughter/AIMS.

**Funding:** This work was supported by the intramural program of the National Institute of Allergy and Infectious Diseases (NIAID), NIH, via grant ZIA AI001076-16 (C.T.B, M.M.S). The funders had no role in study design, data collection and analysis, decision to publish, or preparation of the manuscript.

we outline the capabilities of a new analytical tool, the AIMS: Automated Immune Molecule Separator software, designed for the uniform analysis of all adaptive immune molecules. AIMS accomplishes this cross-receptor compatibility using an amino acid sequence encoding approach that captures key biophysical properties without requiring explicit experimental structural data. The software can be extended to non-immune molecules, making AIMS a widely applicable platform for the broader analysis of protein-protein interactions.

## Introduction

To control infection and disease, the adaptive immune system of higher organisms utilizes a complex collection of receptors and signaling pathways specifically tailored to each individual immunological challenge [1–4]. Over the past decade, researchers have increasingly leveraged these receptors, specifically antibodies and T cell receptors (TCRs), to generate novel therapeutics [5–11]. Generally, the success of natural immune responses or therapeutics are strongly dependent on the ability of these receptors to recognize and appropriately respond to pathogenic threats. However, recognition of pathogens is a dynamic challenge for the immune system as the generation of its receptors is dependent on the identity of the pathogens, and the pathogens themselves are frequently capable of generating compensatory mutations that, in turn, require adaptations in the immune responses.

Both sides of this competition are subject to a balancing act; successful pathogens must mutate and generate variants that reduce detection by the host immune system yet maintain a sufficient level of biological fitness, whereas successful immune responses must recruit or generate receptors that bind with high affinity and specificity to a given pathogen yet ideally maintain sufficient breadth to adapt quickly to these pathogenic variants [12–14]. This biological back and forth is encapsulated by the amino acid sequences that determine the interaction strength between the molecular players involved in adaptive immune recognition. The costs for determining these amino acid sequences of immune receptors has been decreasing rapidly [15], thereby providing access to datasets exponentially increasing in size [16–20]. Likewise, current sequencing technologies allow us to follow the evolution of viruses and identify variants of concern in real-time across the globe [21]. Characterization of the peptides presented by MHC, also referred to as the immunopeptidome, relies on mass spectrometry-based identification. While this method severely limits the coverage of each experiment, single immunopeptidomic assays can yield thousands of identified pathogenic or self-peptides [22]. As these sequence databases continue to expand, methods for analyzing their large datasets must keep pace, helping researchers to identify key distinguishing features of the sequences identified in any given immunological niche.

Excellent software exist for the analysis of TCR sequences [23–28], antibodies [25, 29–32], and peptides [33–35]. Conversely, the analyses of viral sequences are largely dependent on multi-sequence alignments, phylogenetic analysis, or custom pipelines from researchers in a specific viral sub-field. While each of these approaches are powerful tools in their respective fields, they make comparisons across immune repertoires difficult. Software that compares, for instance, peptide and TCR repertoires typically give a simple binary "yes" or "no" to questions of binding, removing the underlying biophysical context that determines these interactions. Further, a majority of the analyses are developed for a very specific task, such as prediction of peptide binding to a specific MHC allele or identification of the evolutionary trajectory of a given antibody sequence. General characterizations of a given immune repertoire

are often done via an in-house analysis, focusing on simplified quantities such as net biophysical properties of sequences, as well as their lengths or conservation.

To facilitate more thorough analyses and comparisons of amino acid sequences, we developed the AIMS (Automated Immune Molecule Separator) software to take into account their fundamental biophysical properties to characterize, differentiate, and identify clusters within immune repertoires. While the initial input and encoding of sequences into AIMS is different for each of the distinct molecular classes of immune repertoires, the downstream analysis is identical and allows for cross-receptor comparisons and the identification of patterns in the corresponding trends of interacting molecules. The application of AIMS for targeted investigations of specific biological systems has been previously described [36–38]. Here, we outline applications of the software to each immune repertoire class with a specific focus on the software's integrated analytical capabilities for cross-repertoire analyses.

## Results

### Encoding amino acid sequences and their biophysical properties

Although the ideal repertoire analysis would build off of complex structures either determined experimentally or predicted computationally, the former approach is inherently low-throughput while the latter is unreliable, even for the most advanced structural prediction software to date [28, 39]. The AIMS software, conversely, takes advantage of the structural conservation inherent to immune molecules, selecting out only the regions involved in the interaction interface. These conserved interacting regions, which are highly variable at the sequence level, are then aligned in matrix form using a pseudo-structural approach that varies across the available analysis modes for each molecular species. By incorporating general structural features, rather than explicit contact predictions, AIMS reduces the bias of the analysis by minimizing the reliance on assumptions of structural accuracy.

Among TCR-peptide-MHC complexes, the interaction interface is strikingly similar, with crystal structures consistently finding nearly identical docking angles between the two [37, 40–44]. TCRs contact the peptide and the MHC $\alpha$-helices via their six complementarity determining region (CDR) loops, which are in turn connected via stem regions to their well conserved framework regions. These stem regions adjacent to the CDR loops are never found within 5 Å of the antigen [26], and are easily identified by highly conserved amino acids, allowing for the exclusion of framework regions from the analysis. In a majority of structures, assuming the conserved stem regions are defined as endpoints, the central 4–5 residues of the CDR3 loops contact the central residues of the peptide [26] (Fig 1A and 1B). From these general structural rules, we can inform the encoding of TCR sequences into AIMS, imposing a "central" alignment scheme as the standard.

In the central alignment scheme, we align each sequence to the central residue of each CDR loop. Whereas most analysis tools segregate TCR sequences by length, thereby artificially segmenting the data, the central alignment scheme of AIMS allows for receptors of all lengths to be analyzed simultaneously while focusing on the key regions of the receptor. Due to the length differences of the TCR sequences in a given dataset, signals from the CDR stem regions will be averaged out, thus prioritizing signals from the center of the CDR loops. We can visualize an example of this encoding for a test dataset of paired TCR$\alpha$ and TCR$\beta$ sequences from the VDJ database [16] (Fig 1C). In this matrix, we can see that each amino acid is encoded as a unique number in the matrix 1 to 21, or a unique color in the figure, with padded zeros between CDR loops represented by white space in the figure. To control for potential artifacts introduced by this approach, analysis can be repeated with "left" or "right" alignment of the sequences, aligning to the N- or C-termini, respectively, of the given sequences (S1 Fig).

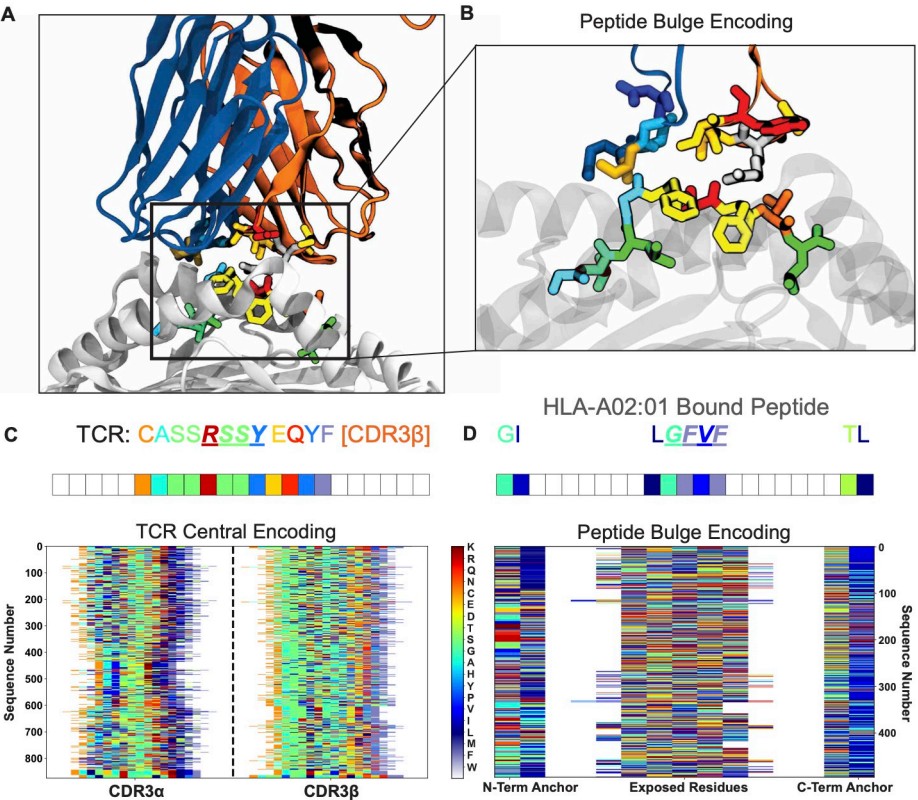

**Fig 1. Example of AIMS encoding for the analysis of TCR-peptide interactions.** (A) Rendering of a specific TCR-pMHC interaction (PDB ID: 1OGA) with TCRα shown in blue, TCRβ in orange, MHC in white. (B) Inset shows a zoom in on this TCR-peptide interface, with the MHC now translucent. Representative AIMS encoding of the single TCR CDR3β sequence (C, central-encoding) or the peptide sequence (D, bulge-encoding) in panel A. Below these single encodings are examples of full TCR repertoire (C) or immunopeptidome (D) AIMS-encoded matrices. Each amino acid in the structures, the single encoded sequences, and the matrices is represented by a unique color.

The standard AIMS encoding for peptides is slightly different from this central TCR alignment scheme. For class I MHC, the flanking regions of bound peptides are frequently 'buried' as highly conserved anchor residues that bind to the MHC platform (Fig 1A and 1B). The majority of TCR contacts are made with the central regions of the peptides that bulge out of the MHC binding groove in the case of longer peptides [45]. However, exceptions to this paradigm may not be uncommon, with TCRs capable of contacting the often-buried peptide N-terminal residue [46] and the C-terminal residue potentially extending out of the MHC pocket [47]. Nonetheless, the length distribution of peptides presented by class I MHC is narrow [33], subverting some of the sequence length concerns present in the TCR analysis. As such, peptide encoding in AIMS adopts a "bulge" scheme. The bulge scheme aligns the N- and C-terminal residues to either edge of the matrix, along with a user-defined number of additional flanking residues. Zeros are padded between these flanking regions and the remaining residues are centrally aligned as in the case of the TCR sequences, again adopting the same numeric amino acid encoding scheme (Fig 1D). We can see clearly for this subset of HLA-A2 presented *Influenza* peptides taken from the Immune Epitope Database (IEDB) [48] the relative conservation at anchor position 2, compared to the variability at the center of the peptide sequences. Importantly, this bulge alignment can also be applied to TCR and antibody sequences, putting more focus on their conserved stem regions.

AIMS is capable of analyzing other molecules with conserved structural features and localized interfacial heterogeneity, including antibodies [36], MHC and MHC-like molecules [37], and, more generally, any molecular subset that can be successfully aligned using existing multi-sequence alignment software [38] (S2 Fig). The generalized AIMS encoding scheme allows for any molecular biologist or bioinformatician to take advantage of the downstream biophysical characterization tools of AIMS for their application of interest. All downstream repertoire characterization follows from this initial encoding, and takes identical paths regardless of the immune repertoire under consideration (S3 Fig). In the following sections we will outline the distinct AIMS modules, applying them to data that best demonstrates the utility of the analyses we perform, rather than opting for a contiguous analysis to a single dataset. More extensive descriptions of AIMS input and output options are provided in the supporting information accompanying this manuscript.

## Unsupervised clustering of a TCR repertoire from an unsorted dataset

To illustrate the implementation of dimensionality reduction and clustering modules in AIMS, we generate a new analysis from the paired-chain data derived from VDJdb [16]. These sequences are complete with metadata regarding their epitope specificity, MHC allele presentation, and the haplotype of the individual each receptor was isolated from, if it was naturally derived. As intuition and recent quantitative work have suggested [49], paired-chain TCR sequence data greatly increases the information content from a given repertoire sequencing experiment when compared to single chain sequencing. Using the dimensionality reduction and clustering modules, we can determine precisely how strongly the analysis changes upon inclusion of the CDR3$\alpha$ sequences (Fig 2).

We first generate the sequence encoding and biophysical property matrices (see S1 Table for list of properties) for both the paired-chain dataset and the CDR3$\beta$-only dataset. From this biophysical property matrix, the sequences are projected onto a three-dimensional space using the uniform manifold approximation and projection (UMAP) dimensionality reduction algorithm [50], and subsequently clustered using the density-based OPTICS (Ordering Points To Identify the Clustering Structure) algorithm [51] (Fig 2A and 2B). It is important to note that, for improved clarity, the projections shown here remove four outlier sequences with a proline in CDR3$\alpha$ and CDR3$\beta$, a very rare amino acid in TCR CDR loops (see S4 Fig for projection with outliers). From the UMAP projection and OPTICS clustering of Fig 2A and 2B we can see that the number of distinct outlier populations is increased in the paired chain dataset, suggesting the identification of a higher number of biophysically distinct sequences.

Visualizing just a subset of the clusters back as encoded matrices (Fig 2C and 2D) we see that this is likely due to the clustering of the paired chain data picking up unique motifs in both CDR3$\alpha$ and CDR3$\beta$, suggesting that these strong outliers can come from either chain. These clusters can then be analyzed for cluster membership compared to their original dataset identifiers, here based on the antigen recognized by each TCR (Fig 2E and 2F). Importantly, cluster purity can be measured relative to the metadata of choice, such as antigenic species source, allele of presenting MHC, organism haplotype, or virtually any other identifiable characteristic that can be encoded as metadata for the sample (S5 Fig).

This visualization of antigen recognition for each cluster (Fig 2E and 2F) highlights subtle differences between the paired-chain and single-chain datasets. We see that despite a nearly equal number of clusters identified, the average cluster purity is higher for the paired chain data at $0.50 \pm 0.28$ when compared to the single chain data at $0.37 \pm 0.24$, although not significantly so. We notice however that the few pure or close to pure clusters are coming from the same two antigens, NLVPMVATV and LLWNGPMAV, largely because these antigens make

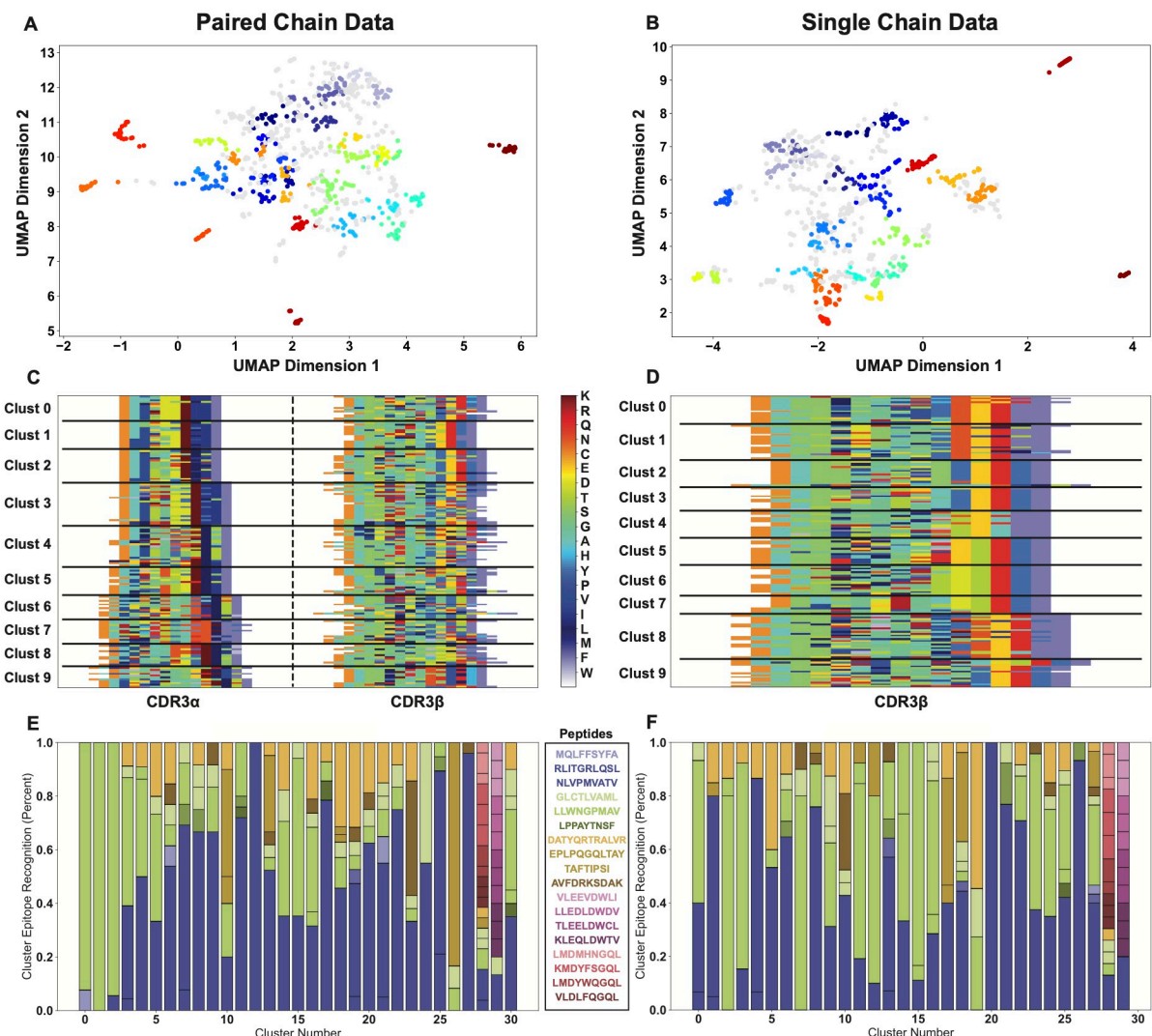

**Fig 2. Comparison of the purity of receptor clustering using paired chain (left) or single chain CDR3B (right) data.** Dimensionality reduction using UMAP, followed by density based OPTICS clustering subsects the data into biophysically similar paired chain (A) and single chain (B) receptors. The first ten of these clusters are then re-visualized in their AIMS-encoded matrix, with black lines marking different clusters (C, D). The antigen specificity of each of these clusters is then quantified by percentages (E, F), with the colors corresponding to specific peptides as shown in the key.

up 43% and 23% of the total dataset, respectively. Overall, these results show that while unique CDR3$\alpha$ motifs are critically important for antigen recognition and for understanding the full breadth of receptor diversity, a fairly accurate picture of sequence diversity and similarity can still be generated from CDR3$\beta$ sequences alone.

## Going beyond receptor clustering and motif analysis

The biophysical analysis in this and the following section can be carried out with individual clusters of sequences derived from unstructured data, as outlined in the previous section, or from antigenically well defined populations (see S6 Fig or Boughter et al. [36] for examples). In highlighting the features of the biophysical property analysis, we select the two antigenically

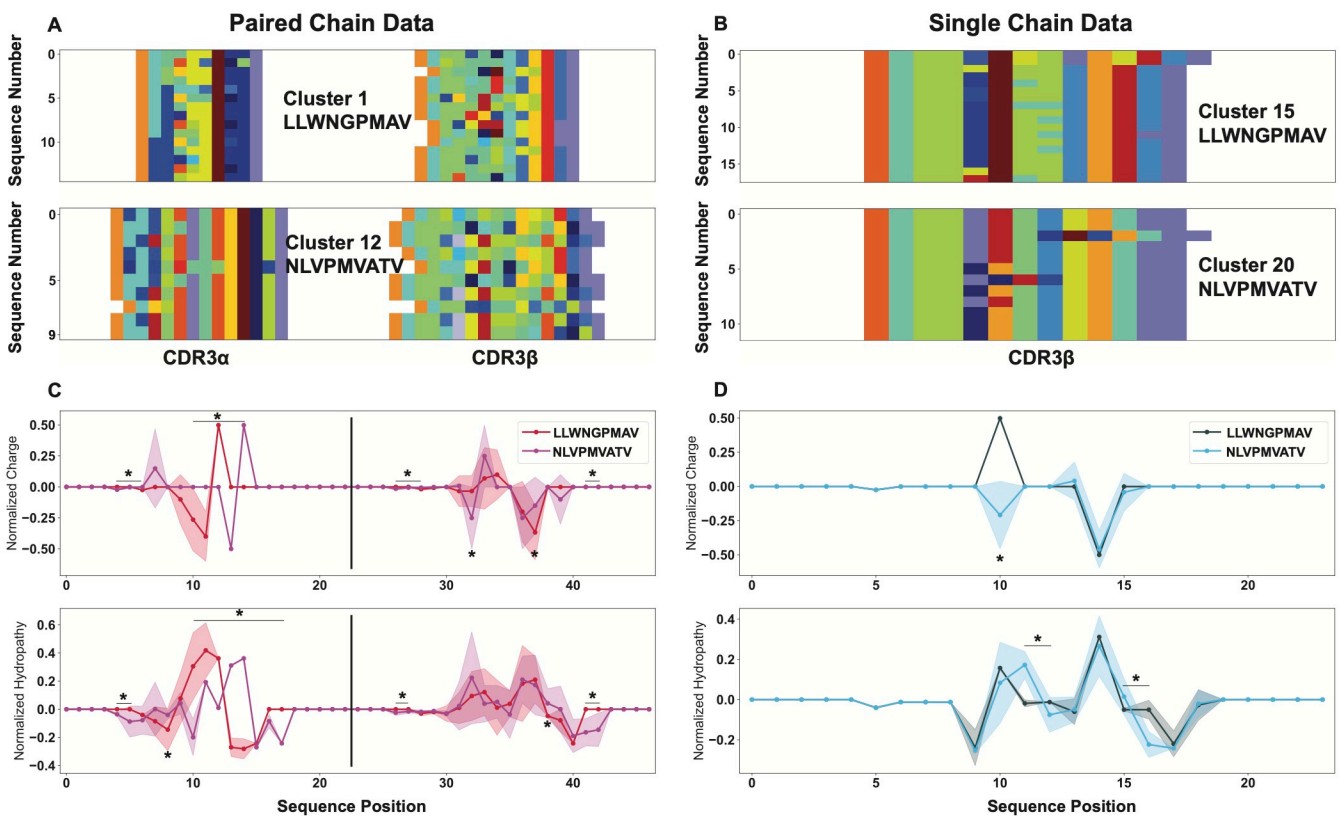

**Fig 3. Isolation of individual sequence clusters and subsequent position-sensitive biophysical characterization of these sequences highlights the details provided by AIMS analysis.** A subset of clusters identified in Fig 2 are isolated and shown as their AIMS matrix encoding for the paired chain (A) and single-chain (B) datasets. From these encodings, we can calculate the position sensitive biophysical properties for each cluster (C, D). Opaque lines with dots represent the averages over each cluster, while wider translucent regions centered on these lines give the variance as calculated by a bootstrapping procedure (Methods). Statistically significant differences ($p < 0.05$, calculated via non-parametric permutation test) are denoted by asterisks, with an asterisk over a solid bar representing extended regions of statistically significant differences.

purest clusters of Fig 2 from both the paired-chain and single chain datasets with specificity to the NLVPMVATV and LLWNGPMAV antigens (Fig 3A and 3B). From these cluster subsets, we can appreciate that despite having nearly the same antigenic purity in each of these clusters, the paired chain clusters surprisingly display increased CDR3$\beta$ diversity. While we are able to generate sequence logos for the distinct motifs in these clusters (S7A and S7B Fig), the AIMS analysis pipeline permits a more thorough biophysical characterization of these clusters.

Specifically, from the high-dimensional biophysical property matrix for each of these clusters, we can isolate single property masks (S7C and S7D Fig) which can be averaged across repertoires (Fig 3C and 3D) or across repertoires and positions (S7E and S7F Fig), generating position sensitive or net average biophysical properties for the molecular subsets of interest. These biophysical property visualizations can be used to more carefully compare and contrast the clusters generated from the paired-chain and single chain datasets. We see from the position-sensitive biophysical property averages that the physical properties of CDR3$\beta$ are matched in the paired-chain and the single-chain clusters, despite the significant difference in diversity in this region. The positively charged segment of CDR3$\beta$ in TCRs recognizing the LLWNGPMAV peptide is seen in both datasets, while a corresponding negative segment is found in the CDR3$\beta$ of TCRs recognizing the NLVPMVATV peptide.

Given the lack of charged residues in the peptides, the conservation of charge in these pure clusters is somewhat surprising. Conversely, the hydropathy score of the CDR3$\beta$ chains is far more variable within the paired-chain clusters, although the same general trend is again followed when compared to the single-chain data, with two distinct peaks in hydropathy and a region neither lacking nor enriched in hydrophobic residues. This is a common feature in the clustering of sequences in AIMS, and may reflect the nature of the hydrophobicity metrics used, or alternatively could be suggestive of the critical determinants of the formation of TCR-pMHC complexes. While charges in the interface must be delicately arranged to form interfacial interactions and offset the significant energetic penalties of desolvation, more hydrophobic residues are free to effectively "fill in the blanks" and pack as best they can. While only the position sensitive charge and hydropathy are highlighted in this section, any of the 61 standard AIMS properties can be visualized either as position-sensitive averages across receptors, or as net averages over position- and sequence-space.

## Generating quantitative metrics of repertoire diversity and amino acid patterning

While the biophysical characterization of immune repertoires can help answer questions regarding the mechanisms of immune recognition, complementary analysis can help contextualize these findings, relating biophysical properties back to patterns in the data resulting from the source of the molecules (e.g. human, mouse, or virus) or other external influences (e.g. selection or affinity maturation). In AIMS, we can further quantify entire repertoires, cluster molecular subsets, or define antigenic groups using an array of statistical and information theoretic approaches. Information theory is built for the analysis of sequences of inputs and outputs [52]. Here, these inputs and outputs are the amino acids making up our immune repertoires quantified via the observed probability distributions across these sequences. In telecommunication, quantification of inputs and outputs determine the messages which can be sent over a given channel, whereas in the study of immune receptors this same quantification determines the range of pathogenic targets which can be recognized by a given immune system. To illustrate this we switch from the VDJdb example to two subsets of isolated peptides from the IEDB: *Influenza A-* and *Ebolavirus*-derived peptides presented by HLA-A*02:01 and HLA-B*15:01, respectively [48]. Importantly, this analysis proceeds without an initial clustering step, and direct differences between the datasets are interrogated as-is.

Our approach leverages the inherent position-sensitivity of the encoded information to build up a site-specific probability distribution (Fig 4A) for these peptide inputs. We see from these position-sensitive probability distributions that the expected anchors at P2 and the C-terminal positions (or P1 and P14 in the AIMS encoding of S8 Fig) have the strongest amino acid preference, as expected for these positions [53, 54]. We see in comparing these two datasets that only P2 leucine (20% enrichment) and P2 glutamine (16% enrichment) show up as distinct anchors for HLA-A*02 and HLA-B*15, respectively, because other strong anchors are shared between these two alleles. Due to the limited overlap in the C-terminal anchors, we see strong preferences towards HLA-B*15 peptides containing PΩ tyrosine and PΩ phenylalanine (15% and 22% enrichments), and a similar preference towards PΩ isoleucine and PΩ valine anchors for HLA-A*02 (21% and 20% enrichments). However, whereas conventional sequence logo plots can generate similar inferences, albeit via a more indirect comparison, our analysis takes these probability distributions a step further. First, from the position-sensitive probability distribution, we can calculate information theoretic metrics such as the Shannon entropy and the mutual information to quantify diversity and relationships among the peptide preferences for these repertoires.

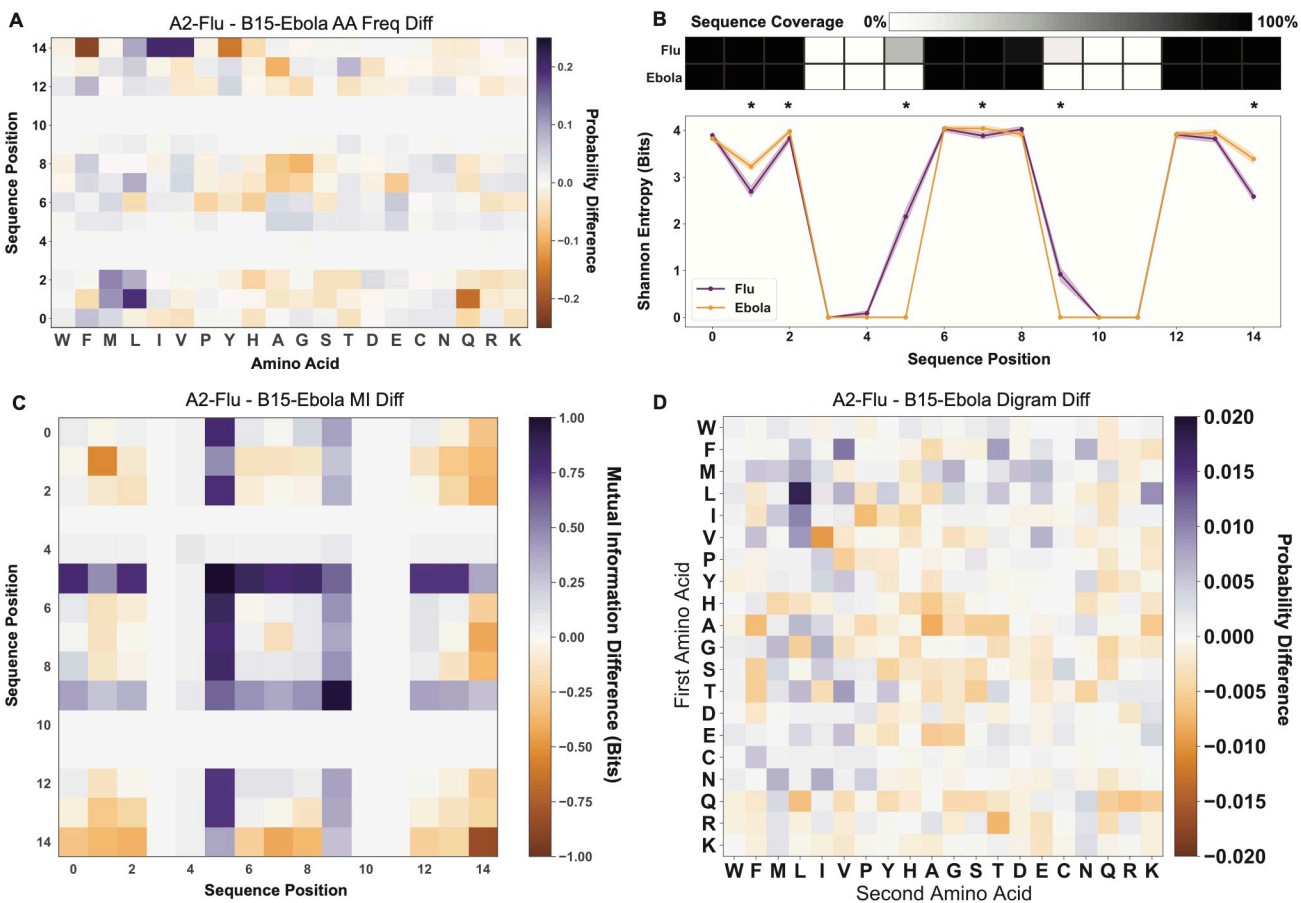

**Fig 4. Statistical and information theoretic AIMS analysis of *Influenza A*- and *Ebolavirus*-derived peptides.** (A) Position sensitive amino acid frequency difference between the two peptide datasets. (B) Position sensitive Shannon entropy quantification and encoding coverage of the two peptide datasets. Statistically significant differences in the entropy ($p < 0.05$, calculated via non-parametric permutation test) are denoted by asterisks. The coverage applies to all position sensitive metrics, and highlights that differences in the entropy and mutual information at positions 5 and 9 are largely due to differences in coverage. (C) Position sensitive mutual information difference between the two peptide datasets. (D) Di-gram amino acid frequency difference calculated between the two peptide datasets. In all difference plots, a deeper shade of purple represents a higher quantity for the HLA-A*02 presented *Influenza A* peptide dataset, while a deeper shade of orange represents a higher quantity for the HLA-B*15 presented *Ebolavirus* peptide dataset. Raw distributions for each individual dataset (S9 Fig) and identification of statistically significant regions (S10 Fig) can be found in the supporting information.

We show the position-sensitive entropy for these two defined peptide populations as well as the sequence coverage resulting from the bulge encoding scheme (Fig 4B). Immediately, we see drops in the entropy in regions of high coverage corresponding to the anchor positions (AIMS P1 and P14). Further, we see that even with peptides derived from single viral clades, the diversity at the center of the peptides is nearly maximal, i.e. all 20 amino acids occurring at nearly equal probability. Given the arguments for the necessity of cross-reactivity in T cells [55], this massive diversity from singular datasets is perhaps unsurprising.

The mutual information is a quantification of the decrease in uncertainty resulting from a known condition. In the case of amino acids it quantifies relationships between amino acid correlations in distinct regions of a sequence. Looking at the difference in the mutual information between the *Influenza A*- and *Ebolavirus*-derived peptides, we see strong trends of increased information between the N- and C-terminal ends of the *Ebolavirus*-derived peptides (Fig 4C). We further see that although only the *Influenza A*-derived peptides have a varied

length distribution, leading to the wider central entropy peak, there is a clear signal in these long peptides, suggesting a specific subset of amino acid usage in the peptides that tend to be longer. Generally in immune repertoire analysis, mutual information may represent instances of co-evolution or receptor cross-talk, as has been discussed in analysis of polyreactive antibody sequences [36]. The goal of this information-theoretic analysis is to identify key regions of increased diversity, conservation, or crosstalk.

As a final step, we can attempt to identify the source of the patterns in the short-range mutual information through the analysis of di-gram amino acid probabilities. Removing the position sensitivity of the peptides in each dataset, we can count the raw occurrence probabilities of each peptide di-gram using a sliding window, building up a probability distribution for each amino acid pair, where the order of occurrence matters (Fig 4D). Interestingly, although there are strong differences in the raw amino acid occurrence probabilities for each dataset (S8 Fig), the di-gram differences are often concentrated in particular regions. For instance, although valine and isoleucine are more frequently found in *Influenza A* dataset, the valine-isoleucine digram is more common in the *Ebolavirus* dataset. In addition to the standard analysis pipeline outlined here, the analysis can be extended to include N-gram motifs, providing the potential to identify regions with a propensity for certain tri-grams or higher-order motifs. Care must be taken when utilizing these N-gram formulations, however, as extension to the extreme such as in the analysis of nine-gram motifs for peptide datasets will identify statistically significant but not particularly meaningful data.

## Comparisons to existing software

While the AIMS analysis pipeline has been developed to address more than just TCR repertoire analysis and clustering, it shares some features with existing software such as GLIPH [26] and TCRdist [24, 56]. These software packages aim to identify TCR sequences enriched above background populations or to cluster sequences with similar amino acid motifs, taking different approaches but generating comparable results to the AIMS clustering outlined above. For GLIPH, we compare our ability to identify distinct motifs using the standard AIMS analysis, whereas in comparing to TCRdist we more quantitatively compare the TCRdist "distance" metric to the corresponding AIMS distance between TCRs.

As a representative motif comparison, we examine the output of GLIPH from Glanville et al. [26] as applied to the *Influenza* antigen M1 presented by HLA-A*02:01 (Fig 5). In addition to the benchmarking of the results of each software, we can compare and contrast the distinct paths to these results. One such difference between AIMS and GLIPH is that there is no "reference population" needed as input for AIMS. AIMS takes the identified M1-reactive sequences (Fig 5A), generates the previously discussed biophysical property matrix, and identifies biophysically distinct clusters in a projected space of these biophysical properties (Fig 5B). Unlike the more diverse clusters of Fig 2 resulting from the analysis of a broad input repertoire, this more targeted analysis of tetramer-sorted sequences results in more homogeneous clustered sequences (Fig 5C). From these clusters, we can then identify the key sequence motifs of each cluster and compare these to the GLIPH results (Fig 5D).

AIMS identifies thirteen biophysically distinct clusters, three of which fully encapsulate the results of GLIPH. We note that while GLIPH identifies the motifs SIRS, IRS, and SIR as distinct, AIMS identifies these sequences in the single cluster SXRS. Further, coloring the sequences by a description of the most basic amino acid properties shows that many of the distinct GLIPH motifs are biophysically degenerate. While the GLIPH results seem to suggest that arginine is a requisite for recognition of the highly hydrophobic peptide GILGFVFTL, we see suggested arginine enrichment is relaxed in the AIMS results, instead suggesting the need

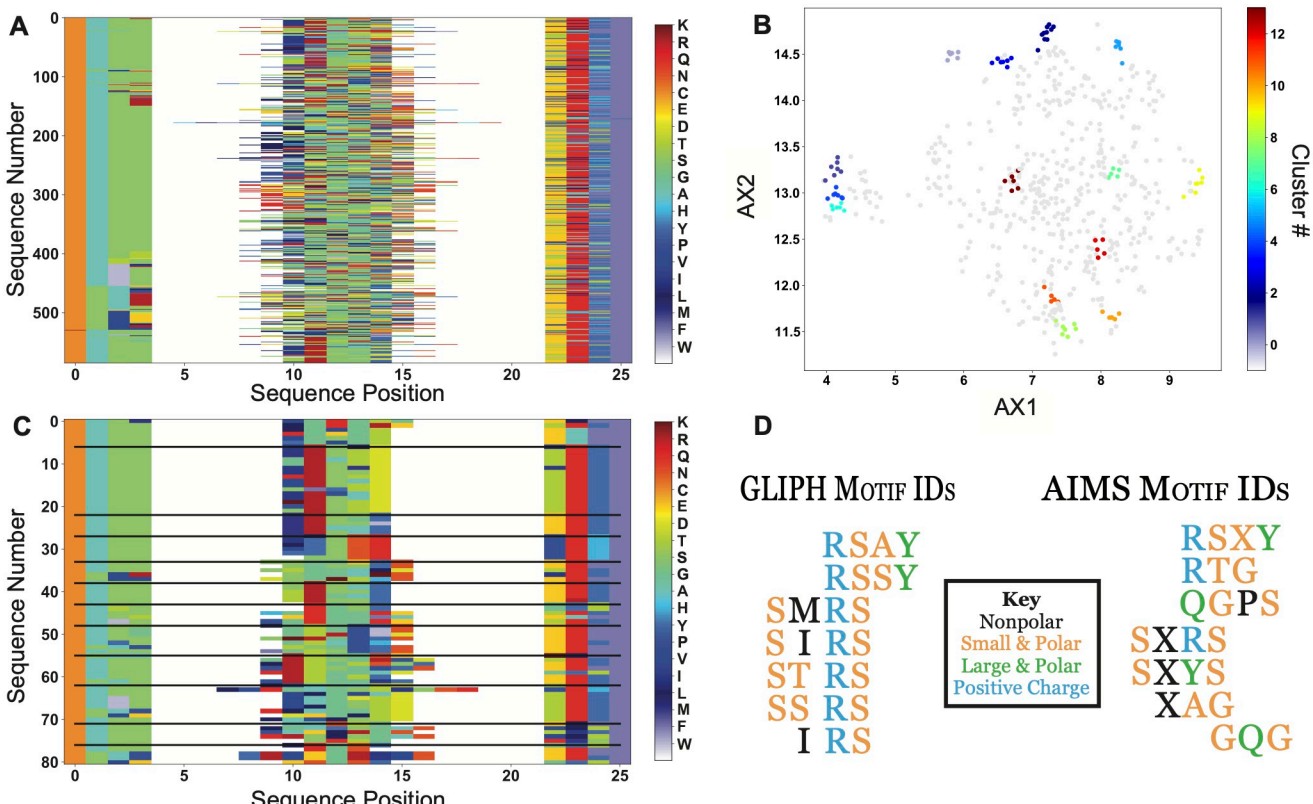

**Fig 5. Comparison of AIMS TCR clustering analysis to GLIPH results.** (A) The *Influenza*-reactive TCRs identified by Glanville et al. [26] are encoded into an AIMS matrix using the bulge-encoding. (B) Each sequence is then processed using the standard AIMS pipeline, and then projected onto two dimensions using UMAP and clustered using the DBSCAN algorithm. (C) These clusters are then re-visualized as an AIMS matrix. (D) Finally, the motifs identified by GLIPH can be directly compared to the motifs identified by AIMS via the clustering in panel C. Biophysical properties of each amino acid in the motif are colored according to the key, and an "X" in the AIMS motif represents "any amino acid with this biophysical property", i.e. the orange "X" can represent S, T, G, or A.

for smaller amino acids (S, T, G, A) or nonpolar residues. Importantly, it is clear that a hydrophilic amino acid is well tolerated or in fact required for recognition, as six of the seven highlighted AIMS clusters have such a conserved residue in CDR3$\beta$.

We can furthermore quantitatively compare the AIMS analysis to TCRdist, an analysis software that clusters and annotates inputs of large TCR repertoires based on similarity. The "distance" metric critical to the TCRdist pipeline provides a useful quantitative comparison point for our AIMS analysis. Importantly, this distance metric is fundamentally based upon the BLOSUM62 substitution matrix [57]. The BLOSUM62 substitution matrix implicitly encodes biophysical similarities and differences between amino acids, whereas the AIMS encoding explicitly encodes all of these biophysical properties in a single high-dimensional matrix. As such, we should expect similar, but not necessarily identical results when calculating the TCR distances using both TCRdist and AIMS.

Using the mouse TCR repertoire data of Dash et al. [24], we first generated the AIMS encodings for either the CDR3 loops of the sequenced TCRs (Fig 6A) or for all six CDR loops of these same TCRs (Fig 6B) with the intention of matching the two main distance output options in TCRdist. We next generated the high-dimensional biophysical property matrices for each of these AIMS-encoded matrices, normalized each feature as discussed previously, and removed highly correlated vectors from each matrix. While the next step in the standard

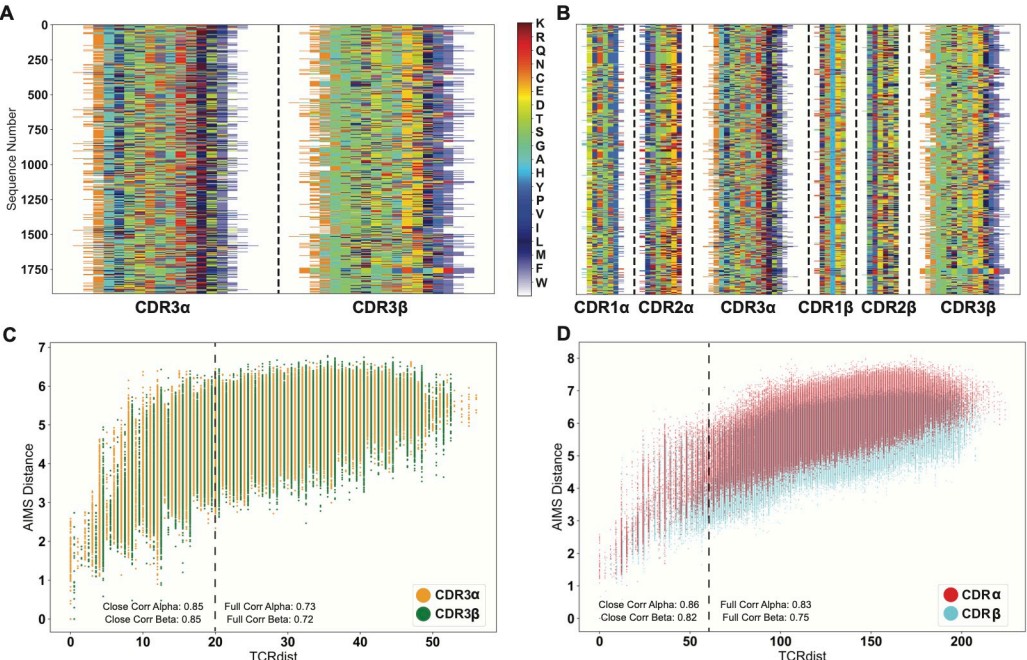

**Fig 6. Quantitative comparison of distance metrics used in AIMS and TCRdist.** Both the CDR3-only sequences (A) and the full-CDR sequences (B) of Dash et al. [24] are encoded into AIMS matrices. Sequence distances calculated via TCRdist and AIMS are then directly compared for these CDR3-only sequences (C) or full-CDR sequences (D) for the TCRα- and β-chains. Correlation coefficients between these distance metrics are reported for the full set of sequences and for closely related sequences, which are delineated by the dashed vertical lines.

AIMS analysis pipeline is the reduction of these high dimensional matrices, the UMAP and PCA projections do not conserve the distances between points. As such, we instead calculate the raw Euclidean distances in the high-dimensional space, and directly compare these to the TCRdist values.

From these panels, we see strong correlation for the CDR3-only distances (Fig 6C, $\rho_\alpha = 0.73$ and $\rho_\beta = 0.72$) and a stronger correlation for the full CDR loop calculations (Fig 6D, $\rho_\alpha = 0.83$ and $\rho_\beta = 0.75$). In both datasets, we see that as the TCRs get more dissimilar, the AIMS distances seem to level off even as the TCRdist metric continues to get larger. This may be due in part to well known issues with the calculation of distance metrics in high-dimensional spaces [58], but is likely not a point of concern in the standard AIMS analysis as clusters are generated in the projected spaces (i.e. in the UMAP or PCA projection). In recent applications of TCRdist [56], a distance cutoff of 20 has been used to define similar CDR3 loops of TCRs. Using this same definition and then a distance cutoff of 60 for all six CDR loop distances, we see that the correlation coefficients improve for more similar TCRs, with CDR3 (all CDR) correlation coefficients of 0.85 (0.86) for the α-chain and 0.85 (0.82) for the β chain. Comparisons with a wider range of antigens from Mayer-Blackwell et al. [56] show similar agreement (S11 Fig).

These new AIMSdist metrics directly inspired by TCRdist can then be used to more quantitatively assess the comparison between AIMS and GLIPH (S12 Fig), highlighting that AIMS is able to recapitulate these results and provide added insights into more biophysically distinct sequence clusters. Thus, in searching through the high-dimensional biophysical space for the same distinct repertoire clusters as GLIPH and TCRdist, AIMS provides not just the identification of these clusters but also the biophysical similarities within these clusters; effectively

providing the explanations for why sequences are grouped into the same clusters. Further, comparisons between clusters or even separate datasets can be made at a level deeper than the motif scale. In highlighting the functionality and unique strengths of AIMS, we set out to show that previous results from TCRdist and GLIPH can be reproduced, and that these results can subsequently be expanded upon with new analyses.

## Discussion

Immune molecules and the pathogens which they protect against represent a unique case study for bioinformatic analysis approaches. The polymorphic regions of TCRs, antibodies, and MHC molecules in humans are concentrated in select regions of these proteins, specifically in their key intermolecular interaction sites, while the rest of their three-dimensional structures are exceptionally well conserved. This structural conservation and localized variability appears to point towards molecular modeling as a key tool, but modern computational approaches are either too costly i.e., slow and inefficient, or too inaccurate to allow for detailed conclusions regarding the proximity of specific amino acids [28, 39, 59]. Many of the best performing machine learning approaches are "black box" algorithms that do not allow the user to determine how or why certain classifications are made. While more interpretable structural modeling approaches are capable of approximate placement of TCR protein backbones on the pMHC surface, even the best structure prediction software struggles to place receptor side chains in the proper position on the antigenic surface [28, 39]. Proper placement of these side chains, to angstrom-level precision, is key for proper inference of the interaction strength between receptors and their cognate antigens.

AIMS was developed specifically as an interpretable tool for immune repertoire analysis, capable of utilizing the information provided by conserved structural features of immune molecules for unique analytical approaches while reducing the potential for errors arising from insufficient resolution offered by structural predictions. The generality of the AIMS analysis pipeline allows for the simultaneous characterization of all adaptive immune molecules, including peptides, conserved viral structures, antibodies [36], MHC molecules [37], T cell receptors, and broadly any protein subset with structurally conserved features and localized diversity [38]. Currently the standard AIMS pipeline analyzes each of these molecular subsets individually, and subsequently allows users to compare and contrast biophysical features of paired TCR-pMHC or antibody-antigen interactions. Recently developed features and further ongoing work is aimed at incorporating mixed repertoire analysis from the pipeline initialization. These features will provide insights into the intricacies and inter-relationships of all of the molecular players involved in an adaptive immune response.

Of the existing software packages available for the analysis of molecular subsets of adaptive immunity, T cell receptor repertoire analysis is the most mature of the fields, and therefore serves as the primary point of comparison for AIMS. Despite not being developed explicitly for T cell receptor analysis, we find that AIMS can reproduce the results of two of the most popular programs developed for TCR analysis, namely GLIPH and TCRdist [24, 26, 56]. GLIPH and TCRdist both utilize distance based metrics to identify unique motifs from TCR repertoire datasets. Extending sequence analysis beyond clustering, AIMS quantifies the biophysical differences between identified clusters of sequences to determine the various approaches TCRs take to recognize single antigenic targets. In the analysis of large experimental datasets, users have the option of clustering using GLIPH or TCRdist, and subsequently importing these sequences into AIMS for the downstream analysis, providing an opportunity to extend the analysis from where other approaches conclude.

One key AIMS approach not discussed in this manuscript is a more targeted, supervised learning approach that is also available for the identification of specific differences in distinct repertoire datasets. This supervised learning approach utilizes a linear discriminant analysis (LDA)-based classifier to simultaneously sort individual sequences into their respective classes and identify key features that delineate distinct repertoires. While other algorithms may perform better in classifying repertoire data, the strength of LDA lies in its interpretability. Unlike other machine learning methods, the vectors most important for discriminating between distinct repertoires are included as output, along with their associated linear weights. Further details on linear discriminant analysis can be found in the extended methods. In this targeted supervised approach, users immediately identify key position-sensitive differences between their datasets.

AIMS fundamentally is built around the identification of biophysically distinct molecular subsets. Independent of any biases in the experimental approaches used, AIMS aims to find the receptor subsets that most strongly span the biophysical space of the input dataset. While still susceptible to some of the issues that plague all analysis in the regime of strong undersampling, by focusing on those receptors that are the most biophysically distinct, rather than those that are the most similar according to a variety of distance metrics, we can identify the limits of molecular recognition. This may under-emphasize the importance of convergence towards certain motifs in immune responses, but it enhances our understanding of the diverse paths adaptive immune systems take towards solving the same problems of pathogenic recognition.

It is important to note that the analytical tools provided by AIMS are best utilized both as a means of identifying differences between datasets and as an exploratory tool. Many of the decisions made at each stage of the analysis can alter the downstream interpretations, so users are encouraged to test different alignments, projection methods, clustering algorithms, and clustering options. While choices in alignment do not strongly alter the underlying structure of the input dataset (S12 Fig and S2 Table), they can generate differential emphasis on distinct features. Further, as seen in S4 Fig, the inclusion or exclusion of certain sequences can distort the projected spaces used for sequence clustering. Thorough investigation should include multiple iterations of analysis, testing how single or paired chain data alter outputs, and how inclusion of mixed or single antigenic specificities in a given AIMS run can provide new and exciting insights. Importantly, AIMS can be easily extended by adding code with the desired additional functionalities, building on the algorithms already present in the software.

## Materials and methods

This section serves as a reference for reproducing the analyses used to create the figures in this manuscript. For a more conceptual overview, readers are referred to the Extended Methods section. For a more practical discussion of the implementation of AIMS using either the graphical user interface (GUI) for non-computational researchers or the Jupyter notebooks and command line interface (CLI) for more advanced users, we direct the reader to download the code through GitHub [https://github.com/ctboughter/AIMS] and follow the walkthroughs available at [https://aims-doc.readthedocs.io/en/latest/].

### AIMS encoding

All sequencing data is first processed into an AIMS-readable format, a simple comma-separated value file with each column corresponding to each structural feature (CDR loops for TCRs, $\alpha$-helices and $\beta$-strands for MHC, or specific regions of interest for multi-sequence alignments). These files are then read into AIMS, parsing out sequences with missing residues, improper characters, or fewer structural features of interest than those defined by the user.

The sequences are then aligned according to user input. For datasets with multiple structural features, alignment is performed independently on each feature. Only "Central" and "Bulge" alignment strategies require special consideration, as both deal with an alignment to a sequence center. The "center" of sequences with an even number of amino acids is chosen to be the amino acid preceding the midway point of the sequence. "Bulge" alignments require additional input specifying the total number of amino acids "padded" on either side of the centrally aligned region. A pad length of 6 (3 AA of the N- and C-termini) was used for peptide analysis of Fig 4, while a pad length of 8 (4 AA of the N- and C- termini) was used for Fig 5.

### Generation of biophysical property matrices

The initial AIMS encoding is used as the template for all downstream analysis. If the sequence encoding matrix utilizes a bulge scheme, then all resultant position-sensitive figures adopt this same alignment. This is accomplished via a simple dictionary, where each amino acid is associated with 62 other values (1 value for positional encoding visualization and 61 for biophysical properties) to generate an i x j x k property matrix. Importantly, a Z-score normalization transforms each individual biophysical property in our dictionary, not the given dataset. Propensities for biophysical interactions can optionally be scored using the pairwise interaction scores of S3 Table. In the analysis throughout this manuscript, vectors with a correlation coefficient above 0.75 with another vector, and all empty vectors (i.e. those corresponding to white space in the positional matrices) are dropped from the biophysical property matrix. Biophysical property measurements (like position sensitive charge, net hydrophobicity, etc.) utilize the full, non-parsed matrices, whereas all projection and clustering is done on these parsed matrices.

### Dimensionality reduction and unsupervised clustering

Dimensionality reduction modules to collapse the high-dimensional biophysical property matrices using either principal component analysis (PCA) or uniform manifold approximation and projection (UMAP) utilize the SciKit-learn [60] and UMAP [50] Python packages. The parsed biophysical property matrices discussed in the previous section are first subject to dimensionality reduction with specified parameters of *n_components* = 3 for both UMAP and PCA, *svd_solver* = *full* for PCA, and *n_neighbors* = 25 for UMAP. These parameters are defaults in AIMS but can be changed by the user. For the purposes of reproducibility, the UMAP random seed was set to 617 for all projections in this manuscript. A discussion of reproducibility when using UMAP can be found within both the AIMS and UMAP Read the Docs pages. All other parameters are SciKit-Learn defaults.

The outputs from these projection algorithms are then fed into either the OPTICS (ordering points to identify the clustering structure) [51] or DBSCAN (density-based spatial clustering of applications with noise) [61] algorithms. For all clustering in this manuscript, the default AIMS specified parameters are used; *min_samples* = 10 for OPTICS and *eps* = 0.15 for DBSCAN. All other parameters are SciKit-Learn defaults. It is important to note that these two parameters should typically be the most variable across applications of AIMS, with the "proper" settings varying strongly across different projection algorithms and input datasets.

### Information theoretic calculations

Information theory, a theory classically applied to communication across noisy channels, is incredibly versatile in its applications, with high potential for further applications in immunology [52, 62–66]. In AIMS, we utilize two powerful concepts from information theory, namely Shannon entropy and mutual information.

Shannon entropy, in its simplest form, can be used as a proxy for the diversity in a given input population. This entropy, denoted as H, has the general form:

$$H(X) = -\sum_X p(x) \log_2 p(x) \tag{1}$$

Where $p(x)$ is the occurrence probability of a given event, and $X$ is the set of all events. We can then calculate this entropy at every position across AIMS-encoded matrices, where $X$ is the set of all amino acids, and $p(x)$ is the probability of seeing a specific amino acid at the given position. In other words, we want to determine, for a given site in the AIMS matrix, how much diversity (or entropy) is present. Given there are only 20 amino acids used in naturally derived sequences, we can calculate a theoretical maximum entropy of 4.32 bits, which assumes that every amino acid occurs at a given position with equal probability.

Importantly, from this entropy we can calculate an equally interesting property of the dataset, namely the mutual information. Mutual information is similar, but not identical to, correlation. Whereas correlations are required to be linear, if two amino acids vary in any linked way, this will be reflected as an increase in mutual information.

In AIMS, mutual information $I(X; Y)$ is calculated by subtracting the Shannon entropy described above from the conditional Shannon entropy $H(X|Y)$ at each given position as seen in Eqs 2 and 3:

$$H(X|Y) = -\sum_{y \in Y} p(y) \sum_{x \in X} p(x|y) \log_2 p(x|y) \tag{2}$$

$$I(X; Y) = H(X) - H(X|Y) \tag{3}$$

Putting these equations into words, we are effectively asking how the knowledge of the identity of an amino acid at one site changes the entropy at another site. If the entropy at the "test site" is zero, i.e. $H(X) = 0$, then no matter what we know of the amino acid identity at another site the change in entropy at the test site will still be zero, and therefore the mutual information will be zero. Likewise, if the entropy remains unchanged at this test site despite the knowledge of the amino acid identity at another site, the mutual information will again be zero. There is only a meaningful mutual information between a test site and a given amino acid site if the knowledge of that given amino acid reduces the entropy at the test site. The mutual information cannot be negative, so the reverse situation, i.e. an increase in diversity with the knowledge of amino acid identity at a given site, cannot occur.

## Statistical considerations in AIMS

The position sensitive averages in AIMS in particular are critical for comparing repertoires from distinct sources or subsets of repertoires, and are capable of identifying different modes of recognition for these molecules. Importantly, these averages are difficult to directly compare statistically, as they are not normally distributed due to the discrete nature of the 20 amino acids composing protein sequences. As such, the AIMS standard for plotting properties of these repertoires is to bootstrap a normal distribution of receptor averages, with the bootstrapped average and the bootstrapped standard deviation plotted in the final figure. The bootsrapped distribution is sampled 1000 times as a default. Statistical significance in AIMS is then calculated using either a two-sided nonparametric Studentized bootstrap or a two-sided nonparametric permutation test as outlined in "Bootstrap Methods and Their Application" [67].

In this manuscript, the two-sided nonparametric permutation test was exclusively utilized to calculate statistical significance. Here, the test statistic $z$ is set to a simple difference of

means, and we randomly permute the data into two bins. We then count the number of permutations where the randomly permuted test statistic is greater than or equal to the empirical test statistic. The p-value is then calculated as:

$$p = \frac{1 + \sharp(z^2 \geq z_0^2)}{R + 1} \tag{4}$$

where $z$ is the permuted test statistic and $z_0$ is the empirical test statistic. $R$ is then the number of tested permutations, here 1000, and $\sharp(z^2 \geq z_0^2)$ is the count of permutations where the square of the permuted test statistic is greater than the square of the empirical test statistic. Assorted p-value cutoffs are reported within the figure legends throughout the manuscript.

### Generation of simulated TCR Datasets

To benchmark quantitative comparisons between different implementations of the AIMS analysis and existing software such as GLIPH and TCRdist, we created a new AIMS module for the generation of simulated TCR datasets. Results of these comparisons can be found in S2 Table. These simulated datasets are generated via a random selection of human V- and J-gene segments, followed by a random deletion of 0–2 amino acids from these selected segments. From there, user inputs determine the number of randomly added amino acids and the probability distributions of these amino acids. It is important to note that the V- and J-gene selection probabilities, the deletion probabilities, and the added amino acid probabilities are pseudo-randomly generated, and are not meant to match biological frequencies.

Our test simulated dataset was comprised of 15,000 total receptors, from three discrete simulated datasets of 5,000 receptors with lengths varying from 11–14 amino acids. The three datasets are named after the amino acid insertion distributions they draw from, with all three datasets excluding cysteine and proline from the distribution. The "Random" dataset sets the probability of insertion of all other amino acids to 1/18. The "KRQN" dataset sets the probability of insertion 20-fold more likely for positive charged amino acids K and R and 10-fold more likely for hydrophilic amino acids Q and N. Likewise, the "DEHY" dataset sets the probability of insertion 20-fold more likely for negatively charged amino acids D and E and 10-fold more likely for amphipathic amino acids H and Y. Such strong trends should generate relatively clean separations using any clustering approach.

This list of 15,000 single chain sequences from these three generated datasets are used as the input for each analysis. In AIMS analysis, the "Standard" approach uses the central encoding scheme, vector normalization and an entropy re-weighting of the biophysical property matrix. This matrix is then projected onto 3 UMAP dimensions and clustered using the DBSCAN algorithm. The remaining entries in S2 Table are deviations from this standard, with the analysis name highlighting which step is changed. So "AIMS Left" utilizes the left alignment scheme, while "AIMS PCA-Kmeans" utilizes the PCA for projection of the data and Kmeans for clustering. TCRdist and AIMSdist clusters are determined using a hierarchical clustering approach with a cutoff of 30 TCRdist units and 4 AIMSdist units. GLIPH clusters are defined by the identified statistically significant motifs with a length of 3 or more and 10 or more sequences per motif, as the clustering algorithm did not converge after 48 hours of continuous calculation.

It should be noted that a true metric of "clustering success" is difficult to quantitatively determine. As such we report a range of statistics for each analysis. For instance, while hierarchical clustering of TCRdist and AIMSdist metrics give nearly 100% cluster purity, the large number of clusters (nearly 400 for AIMSdist and over 500 for TCRdist) may make these results

hard to parse. Further, the number of clusters per dataset is unevenly weighted. We note that while AIMS standard analysis cluster purity appears to be low (75%) the majority of the "contaminants" are from the randomly generated dataset, which likely includes sequences enriched in positive or negative charge. In a way, such "impurities" may be desired in the analysis of especially TCRs, as crossreactivity may make it likely that TCRs from distinct datasets have similar biophysical properties. As a final note, GLIPH performs the worst of nearly all analyses, clustering only 28% of the sequences with single sequences belonging to multiple clusters. This is likely because GLIPH is not meant for the analysis of simulated data, making such comparisons inherently unfair.

## Extended methods

### Dimensionality reduction and unsupervised clustering

In the analysis of immune repertoires, and broadly of amino acid sequences, thorough characterization of these sequences requires the generation of a high-dimensional space comprised of assorted descriptive properties. To systematically analyze this high-dimensional space of data, AIMS employs both linear and non-linear dimensionality reduction techniques with extensive flexibility in the application of these techniques given to users.

Often, we recommend starting with the linear dimensionality reduction, principal component analysis (PCA). PCA is a highly interpretable dimensionality reduction technique that projects the data onto linear combinations of the input vectors corresponding to the orthogonal vectors spanning the dimensions of highest variance in the data. PCA is a powerful and interpretable tool for analysis of high-dimensional datasets, as the identified principal components are fundamental linear algebraic properties of the matrix. Due to the linear nature of PCA, the precise biophysical properties used to create the principal components are easily inferred from the data. Unfortunately, in the analysis of immune repertoires, the axes of highest variance are not always those that best separate the key biophysical features in a given dataset. In antibody and TCR data particularly, the vectors of highest variance in the data will generally be in the center of the CDR3 loops. While this frequently also means that CDR3 will be the most distinguishing feature of a given antibody or TCR sequence, this need not always be the case.

If needed, users can instead turn to the nonlinear dimensionality reduction techniques, specifically t-stochastic neighborhood embedding (t-SNE) and uniform manifold approximation projection (UMAP). Fundamentally these nonlinear algorithms attempt to reduce dimensionality while preserving distance between and within clusters of data points in the original input space. Users should become familiar with each algorithm by reading the relevant literature, but certain key features will be discussed here. Perhaps most importantly, both t-SNE and UMAP as implemented in python are inherently stochastic algorithms. This means that if users want reproducible analysis, care must be taken to first specify a specific seed from which the stochastic algorithm will start from. Further, as nonlinear algorithms the resultant projections are not easily interpreted, making the identification of biophysical differences in localized clusters of data points difficult. Notably however, some of the downstream analytical tools in AIMS can help overcome this shortcoming.

Once the data have been projected onto a lower-dimensional space, users must define their clustering algorithm of choice. The default, Kmeans clustering, is the most conceptually straightforward, breaking the data into N clusters, where N is defined by the user. Kmeans clustering is especially useful if users should expect *a priori* some specific number of clusters arise within their data. In using AIMS for more exploratory studies, density-based clustering is recommended instead, using either OPTICS or DBSCAN algorithms. These algorithms are

not biased by a user-defined number of clusters, and instead identify clusters based on localized concentrations of data points. Each algorithm comes with its own advantages and disadvantages, so again users are encouraged to read further on these to inform their analysis. In the body of this manuscript, UMAP is used to reduce the dimensionality, while the OPTICS algorithm is used to cluster the data.

### Linear discriminant analysis

As discussed briefly in the main text, linear discriminant analysis (LDA) is described in greater detail in Boughter et al. [36] and Nandigrami et al. [38]. Briefly, LDA is useful for two distinct purposes, the generation of classifiers to be used in future applications, and the identification of key features that discriminate between two well-defined datasets. Importantly, both of these applications require large amounts of well-defined data to be confident in the results and a discrete number of labels that can be applied to these data. Unlike much of the data used as examples here, which are more heterogeneous or comprised of mixed populations of data. Despite this, here we briefly define how linear discriminant analysis works in AIMS.

LDA is conceptually similar to PCA, in that the data are projected onto axes generated via a linear combination of the input vectors. However, in LDA the classes which each sequence belongs to are added as input, and the identified axes are those which both minimize within-class distance while maximizing distance between classes. Further, unlike PCA, users must be aware of a potential for overfitting. To avoid this, AIMS includes multiple pre-processing steps before the LDA calculation, including the removal of highly correlated vectors in the biophysical property matrices and a range of algorithms for the selection of a subset of key vectors. When the LDA calculation is completed, the key outputs used in AIMS are the linear weights for each input vector. These weights can then be sorted by magnitude to identify the key properties that best discriminate between the input datasets. These properties can then be visualized in AIMS using the standard biophysical property analysis. For the generation of interpretable classifiers using LDA, more advanced knowledge of machine learning is recommended.

### Supporting information

**S1 Fig. Alternate numerical encoding alignment schemes in AIMS using the same repertoire data as Fig 1C.** Each of these schemes are independently applied to individual key structural features by aligning to the (A) N-terminal amino acids, (B) C-terminal amino acids, or (C) "bulge" encoding as discussed in the peptide analysis described in the main text. Here the bulge padding is set to 6, i.e. 3 amino acids padding the N- and C- termini are separated for alignment, and the remaining amino acids are centrally aligned.
(TIFF)

**S2 Fig. Examples of the input flexibility available in AIMS, with molecular structures on the left and AIMS encodings of subsets of these structures on the right.** (A) Antibody encoding of all six CDR loops, structure via Borowska & Boughter et al. [68]. (B) Multiple sequence alignment encoding as discussed in Nandigrami et al. [38]. (C) MHC and MHC-like encoding of the $\alpha$-helices and $\beta$-strands of these related molecules, structures via PDBs: 2XPG, 1ZT4. (D) Multiple sequence alignment encoding of *Influenza* hemagglutinin (HA) protein, structure via PDB: 1RUZ. *Influenza* MSA via 3DFlu [69].
(TIFF)

**S3 Fig. A graphical overview of the standard AIMS analytical pipeline.** (A) Visual representation of the high-dimensional biophysical property matrix. (B) Representative parsed

biophysical property matrix reshaped into two dimensions. (C) Exemplary dimensionality reduction, clustering, and re-visualization of the data in B. (D) Simplified matrix representation of the linear discriminant analysis workflow. Final repertoire characterization steps using (E) biophysical property analysis or (F) information theoretic analysis of this specific example dataset. Here and throughout AIMS outputs, "sequence position" refers to the encoded position in the AIMS alignment matrix. Vertical black lines in panels E, F, delineate core structural features (here, distinct CDR loops). All position-sensitive figures utilize the same AIMS encoding. Lines and colored boxes help guide the reader through the workflow.
(TIFF)

**S4 Fig. VDJdb repertoire dimensionality reduction and clustering analysis including proline-containing outlier sequences.** Shown are the three-dimensional clustering results for the paired chain (A) and single chain (B) data and the two-dimensional projections of these same figures for the paired chain (C) and single chain (D) data. CDR3$\beta$ amino acid sequences of these outliers are highlighted in the center of the figure. Both sequences were confirmed to be the outliers in the paired chain and single-chain data.
(TIFF)

**S5 Fig. Cluster purity quantification for an array of metadata from the VDJ database when comparing paired chain (left column) and single chain (right column) clustering results.** Antigen species source for each cluster member, with a cluster purity of 0.52 ± 0.26 (paired, A) and 0.39 ± 0.24 (single, B). Presenting MHC of each tested epitope for each cluster member, with a cluster purity of 0.68 ± 0.35 (paired, C) and 0.646 ± 0.37 (single, D). Organism haplotype for each cluster member, with a cluster purity of 0.46 ± 0.26 (paired, E) and 0.38 ± 0.22 (single, F). Legends are comprehensive for panels A, B, C, and D but only show a subset of the groups for panels E, F.
(TIFF)

**S6 Fig. Biophysical and information theoretic analysis, as in Figs 3 and 4, for the defined antigenic sequences reactive to either *Influenza* or *EBV* peptides as used in Glanville et al. [26].** (A) Initial AIMS encoding separated by antigenic reactivity, using the central alignment scheme. (B) Example of a biophysical property mask applied to the data in (A), here specifically showing the position- and sequence-sensitive normalized charge of each sequence. (C) Net biophysical properties, i.e. averaged over all positions and all sequences, for each antigen specificity. (D) Position sensitive charge and hydropathy, i.e. averaged over the y-axis of panel B, for each antigen specificity. Information theoretic analysis concludes the characterization of an antigen-specific repertoire, with the position sensitive entropy (E) and mutual information difference (F). Statistical significance of differences between these two populations are calculated for panels C, D, and E using a non-parametric permutation test (Methods). Averages and standard deviations in panels C, D, and E calculated using a bootstrapping procedure (Methods), with standard deviation in panels D and E represented as a shaded region about the solid line averages. ns—not significant, *—$p < 0.05$, solid bar with * above—contiguous region of $p < 0.05$.
(TIFF)

**S7 Fig. Detailed biophysical analysis done in parallel with Fig 3 for the paired chain (left column) and single chain (right column) selected clusters.** (A, B) Sequence logos of the selected clusters of Fig 3A and 3B, as generated by WebLogo [70]. (C, D) Biophysical property masks of charge (top) and hydropathy (bottom) for each cluster of sequences. The position-sensitive biophysical property masks of Fig 3C and 3D are generated by averaging over the y-axis of these plots. (E, F) Net averaged biophysical properties, i.e. averages over the x- and y-

axes of panels C and D, of four out of the sixty-one available AIMS properties for each cluster of sequences. Statistical significance of differences between these two populations are calculated for panels E and F using a non-parametric permutation test (Methods). *—$p < 0.05$, **—$p < 0.025$, ***—$p < 0.01$, ****—$p < 0.001$.
(TIFF)

**S8 Fig. Associated figures for the peptide analysis of Fig 4.** (A) AIMS-encoding using the bulge alignment scheme of the dataset of *Influenza A* and *Ebolavirus* derived peptides. (B) Position-independent amino acid frequencies for the HLA-A2 presented *Influenza A* peptides and the HLA-B15 *Ebolavirus* peptides. Statistical significance of differences between these two populations are calculated for panel B using a non-parametric permutation test (Methods). *—$p < 0.05$.
(TIFF)

**S9 Fig. Raw distributions for the population differences highlighted in Fig 4.** Position-sensitive amino acid probability distributions for (A) *Influenza A* and (B) *Ebolavirus* derived peptides. Position-sensitive mutual information calculated between each encoded sequence position for (C) *Influenza A* and (D) *Ebolavirus* derived peptides. Amino acid digram frequencies for (E) *Influenza A* and (F) *Ebolavirus* derived peptides.
(TIFF)

**S10 Fig. Associated figures for the statistical significance of the peptide analysis of Fig 4.** Statistical significance is shown for the amino acid frequency difference (A), the average Shannon entropy difference (B), the mutual information difference (C), and the digram frequency difference (D). Statistical significance of differences between these two populations are calculated using a non-parametric permutation test (Methods). For all tests, a threshold of $p < 0.05$ is used, denoted by either the solid red line in panel B or the presence of a filled (black) square in the matrices of panels A, C, and D.
(TIFF)

**S11 Fig. Quantitative comparison of distance metrics used in AIMS and TCRdist.** Here only the sequence distances calculated via TCRdist and AIMS between full-CDR sequences of Mayer-Blackwell et al. [56] are compared directly for the TCR$\alpha$- and $\beta$-chains. Correlation coefficients between these distance metrics are reported for the full set of sequences and for closely related sequences, which are delineated by the dashed vertical lines at a TCRdist of 60 units. TCRs are isolated from human T cells in response to each antigen listed above the plots (A-F) or for the full dataset of TCRs (G).
(TIFF)

**S12 Fig. Quantitative comparison of clustering performance of AIMS and GLIPH on the same dataset.** The AIMS clusters (A) are generated from the curated *Influenza A* reactive sequences from Glanville et al. Supplementary Table 1 [26] subject to a UMAP projection and DBSCAN clustering (eps = 0.15) of the biophysical property matrix. while the GLIPH clusters (B) are taken directly from Glanville et al. Supplementary Table 7 [26]. Calculating the AIMS distances between the sequences within the AIMS clustering (C) or the GLIPH clustering (D) shows highly similar patterns for the most similar sequences, suggesting both methods are capable of identifying highly pure clusters, albeit with a higher resolution in AIMS. AIMS additionally identifies highly biophysically distinct yet self-similar clusters (sequences 50–150) compared to the previously identified specificity groups.
(TIFF)

**S13 Fig. Comparison of the effect of different alignment strategies on the AIMS distance using all TCRs in the Glanville et al. dataset [26].** We see that largely the distances are preserved for similar TCRs, yet there is some divergence in the calculated metric at higher distances for comparisons between the bulge- and center-alignments (left) and the left- and center-alignments (right).
(TIFF)

**S1 Table. List of all of biophysical properties used for this study.** For hotspot detecting variables (HS) a simplified form of the description is used. For more in-depth descriptions, the original reference should be used.
(CSV)

**S2 Table. A quantitative comparison of the accuracy of TCR sequence clustering using a simulated dataset between the different modes of AIMS analysis and TCRdist and GLIPH software.** Details of the simulated dataset and the details of the comparisons can be found in the methods. While "purity" may be considered a useful metric of successful clustering, different approaches will yield different types of clustered receptors, so it is unlikely that a "best" approach exists.
(CSV)

**S3 Table. Table used for the second version of the AIMS scoring of pairwise amino acid interactions.** The table attempts to recapitulate the interactions between amino acids at the level of an introductory biochemistry course.
(CSV)

## Acknowledgments

We thank David Margulies, Caitlin Castro, Ryan Duncombe, and Augusta Broughton for insightful comments and discussions.

## Author Contributions

**Conceptualization:** Christopher T. Boughter, Martin Meier-Schellersheim.

**Data curation:** Christopher T. Boughter.

**Formal analysis:** Christopher T. Boughter.

**Funding acquisition:** Martin Meier-Schellersheim.

**Investigation:** Christopher T. Boughter.

**Methodology:** Christopher T. Boughter.

**Project administration:** Martin Meier-Schellersheim.

**Resources:** Martin Meier-Schellersheim.

**Software:** Christopher T. Boughter.

**Supervision:** Martin Meier-Schellersheim.

**Validation:** Christopher T. Boughter.

**Visualization:** Christopher T. Boughter.

**Writing – original draft:** Christopher T. Boughter.

**Writing – review & editing:** Christopher T. Boughter, Martin Meier-Schellersheim.

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
