## [Decision Letter · Decision Letter 0]

8 May 2023

Dear Boughter,

Thank you very much for submitting your manuscript "An Integrated Approach to the Characterization of Immune Repertoires Using AIMS: An Automated Immune Molecule Separator" for consideration at PLOS Computational Biology.

As with all papers reviewed by the journal, your manuscript was reviewed by members of the editorial board and by several independent reviewers. In light of the reviews (below this email), we would like to invite the resubmission of a significantly-revised version that takes into account the reviewers' comments.

Dear authors,

Please accept my sincere apologies for the long delay in the review process.

Both reviewers appreciated the importance of the problem of developing biophysical encodings of immune receptors and their ligands, but also raised major concerns regarding readability, contextualization with regards to prior studies, and statistical validation.

If the authors think that these concerns can be suitably addressed by a major revision of the manuscript, we would be glad to reconsider the manuscript for publication in Plos Computational Biology.

Best wishes,

Andreas Mayer

Guest Editor

We cannot make any decision about publication until we have seen the revised manuscript and your response to the reviewers' comments. Your revised manuscript is also likely to be sent to reviewers for further evaluation.

Sincerely,

Andreas Mayer, Ph. D.

Guest Editor

PLOS Computational Biology

Lucy Houghton

Staff

PLOS Computational Biology

Dear authors,

Please accept my sincere apologies for the long delay in the review process.

Both reviewers appreciated the importance of the problem of developing biophysical encodings of immune receptors and their ligands, but also raised major concerns regarding readability, contextualization with regards to prior studies, and statistical validation.

If the authors think that these concerns can be suitably addressed by a major revision of the manuscript, we would be glad to reconsider the manuscript for publication in Plos Computational Biology.

Best wishes,

Andreas Mayer

Guest Editor

Reviewer's Responses to Questions

**Comments to the Authors:**

Reviewer #1: The authors present an encoding scheme for a more comprehensive analysis of immune repertoires.

However, in the current form, this manuscript doesn't make much sense. The text is a random assortment of statements with motivation or research question and the figures are difficult to make sense of.

I like the idea of a new encoding scheme for immune receptor data that enables novel and more unbiased analyses. However, in the current form, I don't see or understand how the authors achieve this goal.

I suggest the authors rewrite their manuscript with more focus and clarity. Also, the manuscript seems to be TCR-heavy, a bit more space on the antibody side would be nice.

Reviewer #2: This paper provides a useful and important perspective for the sequence analysis of TCRs and antigens presented on MHC. The core of the approach is encoding methodology for sequences. This encoding couples a centered alignment (TCRs) or an anchored + centered alignment (pMHC) along with a high dimensional biophysical representation of each amino acid. Furthermore, the authors have put an impressive effort in making their code accessible, even building a GUI interface – this is commendable in a field that often assumes technical knowhow and rarely puts a premium on usability.

The authors show the usefulness of their method in three main ways:

1) Characterization of biophysical properties of collections of sequences (Figs 2, 4, S5, S6)

2) Clustering of similar sequences (Figs 3, 6, S3, S4)

3) Probabilistic and Information theoretic characterization of collections of sequences (Figs 2, 5, S5, S7)

The characterization of biophysical properties is of particular interest and would be a useful tool if the following points can be addressed.

Major points:

1) The core of AIMS is the encoding of sequences, of which alignment is a crucial part. It would be very helpful if the authors quantitatively compared different alignment schemes. While there are some qualitative explanations for the centered vs bulge alignment schemes, there is no quantitative comparison of downstream analyses (many of which are entirely conditional on position in the alignment) between these strategies and other alignment methods. In particular, how does a standard multi-alignment, or a combination left-right alignment perform? Also, why is the bulge alignment method used for the TCR sequences in Figure 6 (comparison with GLIPH).

2) The authors don’t include discussion of statistical significance or sample size effects throughout the paper. The authors mention that there is a built in tool in AIMS to do bootstrap estimation of errors – they should include this statistical analysis wherever they can. I highlight a few areas in subsequent comments.

3) The alignment strategies used by AIMS means that the coverage of different positions varies. This is never explicitly shown and has both noise effects and direct finite sample size effects on some quantities. For example, the estimate of entropy is capped at log(N) where N is the number of points. You can clearly see the entropy going to 0 in areas that have low coverage. I highlight areas where this is an issue, but plots of the alignment position coverage should be included (Figs 2, 4, 5, S5) and a note made about why the entropy goes to 0 in places. The authors may also want to consider using a normalized entropy.

4) For the clustering module, there is no discussion or analysis on the repeatability/stability of the clustering. TCR repertoire samples can vary wildly in size over a few orders of magnitude, so a sense of how sample size dependent the clustering is and how stable it is to subsampling would be extremely helpful.

5) Figure 5. It would be useful to show the probability distributions, mutual information, and di-grams for the two peptide clusters individually before you show the difference between them. As mentioned in major point #3), using normalized entropy will highlight the dip in entropy due to the conserved anchors. Lastly, there are finite sample size effects on the mutual information (e.g. see Treves and Panzeri https://www.tandfonline.com/doi/abs/10.1080/0954898X.1996.11978656). The authors should check the magnitude of these effects as it is not clear that any of these differences plotted are significant or noise.

6) The authors do not include any results or plots for the machine learning/LDA classifier techniques. Either such an analysis should be added or the section from lines 378-394 should be cut from the results section and the application discussed in the discussion. Similarly, the section in the methods for LDA should be cut unless the analysis is used in the paper.

7) Comparison to GLIPH. The authors have an excellent methodology for characterizing the effectiveness of their own clustering. They should do a comparison between GLIPH clusters and their own using cluster purity (or other quantity – see minor point #2) in order to see which method performs better.

8) Details on how some quantities and plots (e.g. cluster purity, how amino acid gaps are factored into biophysical averages) are missing from the methods. Similarly, any parameter choices for the alignments and clustering should be included.

Minor points:

1) The authors stress the importance of not conditioning the analysis on sequence length, however much of the biophysical and information theoretic analyses performed are done on clusters of sequences with almost identical sequence lengths (e.g. TCR clusters in Fig 4, HLA presented peptides in Fig 5). Does AIMS cluster sequences of similar lengths together? Is the analysis of position dependent biophysical properties dependent on similar length sequences?

2) The authors do not concretely translate what “Position” is on the x- axis for Figs 2E and 2F. It should be made clear if this is position in the aligned TCR sequence space, and furthermore what the overall coverage of sequences per position is (see major point #3). The authors should also explain what the vertical black lines are. It would also be useful to briefly state what the example repertoire being plotted is.

3) The authors use a nice readout of cluster purity to assess the effectiveness of their clustering. This is a very nice analysis, but the quantity is sensitive to the overall distribution of the categories. This is a bit of a problem as there is a significant skew in the datasets used (e.g. HLA-A*02 is far more studied than other HLA and comprise a large fraction of the total peptides). While not essential, the authors may want to use a quantity that isn’t so sensitive to this skew (e.g. an entropy based calculation or even just mutual information).

4) The alignment of MHC class I presented antigens assumes a central bulge. While this is a good assumption for most human HLA some human HLA and mouse MHC have central anchor positions (for example: https://www.ncbi.nlm.nih.gov/pmc/articles/PMC2248166/). Is this bulge model applicable in these circumstances? Is a more flexible approach possible to align the sequences to account for the variability in anchor positions?

5) In the clustering module (lines 237-241), the authors speculate on why TCR 1A, a tumor-isolated TCR, does not cluster. Speculating over a single TCR sequence is a bit of a stretch especially as they do not highlight what specific biophysical property of this TCR makes it an outlier.

6) Fig 4 would be improved by taking coverage into account (see major point #2). I would also recommend aligning the sequence representations in A and B to the position axes in C and D. It would be helpful to the reader to identify the clusters as being specific to the NLVPMVATV antigen or LLWNGPMAV antigen instead of the legends “Clust1” vs “Clust12”. Also, the plots are a bit confusing to follow. Can the authors plot the mean +/- std? At the least choosing colors that are more distinguishable than pink and purple would be helpful -- the highly transparent plots are hard to tell apart.

7) Lines 370-373. It would be useful if the authors had a methodology for determining significance of N-gram motifs, but this may be beyond the scope of this paper.

8) Lines 373-377. This section is written in a confusing manner. It may clarify things for the reader if the authors specified that entropy is additive when the distributions are independent, so the summation provides an upper bound (as they state).

9) The methods around the AIMS software reads like a README. A more concise methods that provides the precise parameter choices and algorithms used to generate the results of the paper would be preferred.

**Have the authors made all data and (if applicable) computational code underlying the findings in their manuscript fully available?**

Reviewer #1: Yes

Reviewer #2: Yes

PLOS authors have the option to publish the peer review history of their article (what does this mean?). If published, this will include your full peer review and any attached files.

Reviewer #1: No

Reviewer #2: No
---

## [Decision Letter · Decision Letter 1]

14 Aug 2023

Dear Boughter,

Thank you very much for resubmitting your manuscript "An Integrated Approach to the Characterization of Immune Repertoires Using AIMS: An Automated Immune Molecule Separator" for consideration at PLOS Computational Biology.

We have sent your revised manuscript back to the two reviewers that reviewed the initial submission. As the previous round of reviews had raised major concerns regarding readability, scientific focus, and statistical rigour, we additionally sought the expert input from two further reviewers on the revised manuscript.

In light of the reviews (below this email), we unfortunately cannot proceed to publication at this time, but invite the resubmission of a significantly-revised version that takes into account the reviewers' comments. Specifically, the reviewers have raised concerns regarding scientific novelty with regards to the prior publication of the AIMS framework, and the lack of clear novel scientific results, which would need to be addressed in a revised manuscript.

We cannot make any decision about publication until we have seen the revised manuscript and your response to the reviewers' comments. Your revised manuscript is also likely to be sent to reviewers for further evaluation.

Sincerely,

Andreas Mayer, Ph. D.

Guest Editor

PLOS Computational Biology

Lucy Houghton & Rob J. de Boer

PLOS Computational Biology

Reviewer's Responses to Questions

**Comments to the Authors:**

Reviewer #1: The authors have addressed all of my concerns.

Reviewer #2: The authors have substantially improved the content of the paper. In particular, the addition of statistical considerations in almost all of the previously flagged areas highlights where the AIMS may be most useful. There are a few (mostly minor!) outstanding issues listed below:

Content revisions:

1) The authors have still not adequately addressed the finite sample size issues for the information theory quantities. While the authors have accounted for the upper bound of the entropy due to the state-space having 21 possible entries, they still do not account explicitly for the varying coverage due to alignment. The maximum entropy for a particular position will also be capped by the coverage: S_max <= log(max(N, 21)) where N is the number of sequences that align to the position without the padding. It is crucial to include a plot alignment coverage by position so that the finite sampling can be accounted for: as the authors note, if a different alignment scheme is used the coverage is different and the resulting entropy by position plots change substantially.

An example of this is Figure 5. While I cannot be sure of the particulars without a coverage plot, the lack of alignment to positions 3-5 and 9-11 is the dominant features of figs 5B and 5C. Furthermore, the coverage differences between these peptide pools probably accounts for some of the most visually obvious features of the plot. Based on the plots I would guess that the HLA-A*02 Flu peptide pool includes more peptides longer than 9 amino acids so the alignment more often includes positions 5 and 9 which appear to never be aligned to by the HLA-B15 Ebola peptide pool (again, without a coverage plot this is impossible for me to determine). If a reader is not aware of these coverage artifacts they will likely be focused on that instead of the likely more biologically relevant entropy reduction at the anchor residues.

I would suggest the following actions to rectify this:

a) At a minimum, coverage plots of the alignment of each peptide pool should be included in the Supplementary figures

b) The finite sample size effects of entropy could be largely dealt with by only analyzing positions for which the coverage is sufficiently high (i.e. >20). Similarly it is probably wise to restrict the position-dependent amino acid probability distributions to positions with sufficient coverage.

2) If the authors have decided to not include any LDA analysis the discussion of LDA applications should be restricted to the discussion, not the results section (see paragraph from lines 167-180) or a main figure.

3) Figure 4 is much improved! The addition of the regions of significance is a great addition.

I would recommend making sure the yticks in fig 4A are integers. For C/D) I still suggest adding an xlabel, but the figure alignments with A/B do alleviate the main concern.

4) While GLIPH is normally used as a qualitative motif tool, I would just reiterate that I think the authors are missing an opportunity for a quantitative comparison of the clustering functionality. This may not be essential for publication, but the authors have an opportunity to demonstrate to potential users any advantages they have over GLIPH groupings.

Style concerns:

While I understand that the authors want to highlight the flexibility and motivation of each element of the AIMS tool, the result is that this paper is long, wordy, and reads like a README with the authors addressing the “AIMS user” instead of the reader. I flag this not as a concern for publication, but instead for the potential reach of AIMS as a tool. I think the utility of the tool would be better highlighted if the text was half the length; with some of the exploratory/”user options” cut; and more of a focus on what can be learned from the results. The authors don’t want to miss out on potential users of their tool because readers cannot get through the paper.

Reviewer #3: This is a description of the AIMS software package, meant to complement the original publication and present some new features. The manuscript is clearly written. There don't seem to be any specific novel biological insights; the manuscript is more a series of usage examples. Here are a few specific comments and suggestions.

Line 112: "the flanking regions of bound peptides are ‘buried’ as highly conserved anchor residues that bind to the MHC platform and are unable to contact TCR " -- this is not true for peptide position 1, which points outward and can be contacted by the TCR and serve as a specificity determinant (talking about MHC class I; see e.g. PDB 5jzi).

Could the amino acid colors in Fig.1 be made more consistent with biophysical similarity, rather than just based on alphabetical ordering? Following some of the previous sequence-logo type color schemes, for example, where positively charged amino acids are blue, negatively charged are red, polar are green, nonpolar gray (or variants of these colors). It might make the alignments more visually interpretable. It doesn't make sense to me that K and M are right next to each other (and A and R), for example, or that K and R are so far away. A similar argument could be made for ordering the amino acids by similarity, in figure 5A for example, rather than alphabetically. This might make it easier to see trends in preferences. Something like: WYFMLIVAPGCSTNQDEHRK

How are the property values for missing/gapped sequence positions handled? Are they assigned the mean value?

The legend is messed up in Fig 4D

Figure 5A-- doesn't convey much information to me; does not seem to match the text description. "only P2 glutamine and isoleucine show up as distinct anchors for HLA-B*15 and HLA-A*02, respectively" Actually it looks like leucine and methionine have stronger relative preferences (darker blue) than isoleucine at P2.

Figure 5B-- the handling of gaps makes this a very confusing plot; shouldn't the anchor positions be the lowest entropy positions? Instead we have these artificial 0 values suggestive of high conservation but they are just positions without any amino acids. Or worse, intermediate but still very low entropy values (pos 5) from a mix of gaps and non-gaps. This same issue extends to 5C where we have the most striking feature of the plot coming from the difference along the diagonal (ie, just entropy) at this silly position 5 where one set of peptides has all gaps and the other doesn't.

Figure 6: "the motifs identified by AIMS via the clustering in panel C" How, specifically, are motifs identified in the AIMS clusters? I couldn't find a description in the methods. A potential advantage of the GLIPH motifs is that they are nominally statistically significant based on a background repertoire. The AIMS clustering/motif ID procedure will presumably find clusters and motifs, even in TCRs from naive T cells without shared binding specificity. For example in shared V or J sequence regions.

Reviewer #4: This manuscript describes a novel analysis tool for the characterization of immune-related peptides/proteins based on their amino acid sequence. The authors have expanded an analysis pipeline that was previously developed only for antibody repertoire characterization, to now process and characterize essentially any set of amino acid sequences, with a focus on immune-related molecules. To achive this generalized functionality, they apply sophisticated alignment, encoding and clustering approaches.

The analysis of immune-related peptide/protein repertoires is becoming increasingly important with a fast growing number of available datasets. Understanding the diversity and functional properties of peptide/TCR/antibody repertoires is a key requirement for antigen-specific immunotherapy approaches. The aim of this manuscript is therefore of great importance and very timely.

However, while the goal to characterize any ‚immune repertoire‘ is applaudable, it comes with the significant risk to ignore or neglect specific characteristics of certain repertoire types (e.g. peptide vs. protein, MHC vs. TCR). In fact, it remains unclear to me what the specific questions would be that could be addressed with such a generalized tool that can purportedly analyze all sorts of sequences. From the abstract and main text it sounds as if repertoires of interacting peptides and MHC molecules could be magically clustered in order to identify certain peptides bound by certain MHC molecules. However, an MHC molecule doesn’t bind a peptide because it has a similar sequence, so I don’t understand how this should work. The same is true for the interaction of TCR (CDR3) and MHC:peptide complexes, how would the clustering based on amino acid sequence help in understanding their interaction?

As the authors point out, there is already excellent software to characterize each of the different immune repertoire types (antigen/TCRs/MHCs/antibodies). I’m lacking the imagination in which circumstances I would need a tool that can characterize all in one go (and what I could learn from this). The authors state for instance that “Software that compares, for instance, peptide and TCR repertoires typically give a simple binary “yes” or “no” to questions of binding, making the identification of trends within or across these repertoires difficult”. But what kind of trend would I expect across these repertoires? It would be fantastic if this tool could predict which TCR can bind which peptide, but this would be wishful thinking, as this has nothing to do with amino acid sequence similarity between peptide and TCR.

Another example is this statement: “[AIMS] allows for cross-receptor analysis and the identification of patterns in the corresponding trends of interacting molecules.” I have no idea what the authors mean with this (‘patterns in the corresponding trends of interacting molecules’).

I’m also concerned by the issue, which became particularly clear through the highly insightful review by reviewer #2, that for many of the possible parameter settings that impact on the analysis outcome are not well described. The authors brush over this by stating that the user should know what they are doing and should try many different settings to see how it affects their results. But what is the user to do if the results change depending on the setting? How are they to know which are the ‘right’ results (and settings). I would argue that the developers of a tool should be the ones to show the user which parameter settings are critical for appropriate analyses and give guidelines what should be used in which case. The authors state that the tool should be used for exploration, but the user needs some guidelines about how different results are to be interpreted, based on thorough testing of known/simulated data.

Beyond the relevance and usability of the new tool, the manuscript describes in great detail the computational steps performed by their pipeline. I have to admit that I lack the computational and mathematical background to judge some of the employed approaches, so I’m not questioning those. However, reviewer #2 seems to have dipped deeply into the methodology and examined/reviewed it carefully, finding it generally sound.

The comparison to existing tools (e.g. GLIPH or TCRdist for TCR clustering) is only partly helpful, since it is difficult to interpret the differences to their results. Which tool is right when there are differences? Here it would be helpful to have a simulated test dataset where we know what to expect in terms of output.

Overall, it seems to me that the authors are here describing an advanced amino acid sequence clustering tool that might be useful also in the context of immune repertoire analysis, e.g. for the analysis of a TCR sequence dataset. If it does better in this clustering than other existing tools remains unclear to me. However, I find the current framing with cross-repertoire analyses highly confusing and even misleading. What is the biological meaning of comparing amino acid sequences of TCRs with those of peptides?

**Have the authors made all data and (if applicable) computational code underlying the findings in their manuscript fully available?**

Reviewer #1: Yes

Reviewer #2: Yes

Reviewer #3: Yes

Reviewer #4: Yes

PLOS authors have the option to publish the peer review history of their article (what does this mean?). If published, this will include your full peer review and any attached files.

Reviewer #1: No

Reviewer #2: No

Reviewer #3: No

Reviewer #4: No
---

## [Decision Letter · Decision Letter 2]

6 Oct 2023

Dear Boughter,

We are pleased to inform you that your manuscript 'An Integrated Approach to the Characterization of Immune Repertoires Using AIMS: An Automated Immune Molecule Separator' has been provisionally accepted for publication in PLOS Computational Biology.

Let me also take this time to apologize for the long review process. This was a complicated paper to find suitable reviewers for due to its somewhat unorthodox structure as a companion paper to a computational software tool. Thank you for your patience -- I think the paper is better for your additional work and rewriting prompted by the reviewer comments, but I appreciate that the process has lasted for a long time.

Best regards,

Andreas Mayer, Ph. D.

Guest Editor

PLOS Computational Biology

Lucy Houghton

%CORR_ED_EDITOR_ROLE%

PLOS Computational Biology

Reviewer's Responses to Questions

**Comments to the Authors:**

Reviewer #3: My concerns have been addressed.

**Have the authors made all data and (if applicable) computational code underlying the findings in their manuscript fully available?**

Reviewer #3: Yes

PLOS authors have the option to publish the peer review history of their article (what does this mean?). If published, this will include your full peer review and any attached files.

Reviewer #3: No

---

## [Editor Report · Acceptance letter]

16 Oct 2023

PCOMPBIOL-D-22-01872R2 

An Integrated Approach to the Characterization of Immune Repertoires Using AIMS: An Automated Immune Molecule Separator

Dear Dr Boughter,

I am pleased to inform you that your manuscript has been formally accepted for publication in PLOS Computational Biology. Your manuscript is now with our production department and you will be notified of the publication date in due course.

With kind regards,

Dorothy Lannert
